# The Structure Properties of Carbon Materials Formed in 2,4,6-Triamino-1,3,5-Trinitrobenzene Detonation: A Theoretical Insight for Nucleation of Diamond-like Carbon

**DOI:** 10.3390/ijms241612568

**Published:** 2023-08-08

**Authors:** Zheng-Hua He, Yao-Yao Huang, Guang-Fu Ji, Jun Chen, Qiang Wu

**Affiliations:** 1National Key Laboratory of Shock Wave and Detonation Physics, Institute of Fluid Physics, China Academy of Engineering Physics, Mianyang 621900, China; herary-hezhh@caep.cn (Z.-H.H.);; 2National Key Laboratory of Computational Physics, Institute of Applied Physics and Computational Mathematics, Beijing 100088, China

**Keywords:** carbon sp^2^- to sp^3^-hybrid, diamond-like nucleus, microscopic structure properties, nucleation mechanism, molecular dynamics simulation

## Abstract

The structure and properties of nano-carbon materials formed in explosives detonation are always a challenge, not only for the designing and manufacturing of these materials but also for clearly understanding the detonation performance of explosives. Herein, we study the dynamic evolution process of condensed-phase carbon involved in 2,4,6-Triamino-1,3,5-trinitrobenzene (TATB) detonation using the quantum-based molecular dynamics method. Various carbon structures such as, graphene-like, diamond-like, and “diaphite”, are obtained under different pressures. The transition from a C sp^2^- to a sp^3^-hybrid, driven by the conversion of a hexatomic to a non-hexatomic ring, is detected under high pressure. A tightly bound nucleation mechanism for diamond-like carbon dominated by a graphene-like carbon layer is uncovered. The graphene-like layer is readily constructed at the early stage, which would connect with surrounding carbon atoms or fragments to form the tetrahedral structure, with a high fraction of sp^3^-hybridized carbon. After that, the deformed carbon layers further coalesce with each other by bonding between carbon atoms within the five-member ring, to form the diamond-like nucleus. The complex “diaphite” configuration is detected during the diamond-like carbon nucleation, which illustrates that the nucleation and growth of detonation nano-diamond would accompany the intergrowth of graphene-like layers.

## 1. Introduction

Carbon has various allotropes owing to its diverse bonding properties with different orbital hybridizations, resulting in extensively distinct physical features [1,2]. Among them, the nano-diamond is the most fascinating for its outstanding properties in hardness, biocompatibility, optics, and electronics, being widely applied in various fields [3,4], such as medicine and sensors. However, the synthesis and structure controlling are far away from practical demand. Since the nano-diamond was detected in detonation soot (called detonation nano-diamond (DND)) of (2,4,6-trinitrotoluene) TNT mixture explosives more than 30 years ago [5], a new avenue was opened for the synthesis of nano-diamond particles. Numerous efforts have been devoted to studying the formation of DND with different explosives and detonation conditions [6,7,8,9,10,11,12,13,14,15], but the nucleation and aggregation mechanism is still an issue with extensive controversy [16].

The detonation nano-diamond is well known as an aggregation, consisting of main nanometer particles surrounded by a matrix of amorphous carbon [17]. Thus, it is believed that the nucleation and growth of DND would be a bound aggregation process rather than free agglomerate [18], which is intrinsically different from the well-understood mechanisms of the biased enhanced nucleation (BEN) [19] and chemical vapor deposition (CVD) [20]. The extreme detonation condition, coupled with flexible bonding ability, can bring several probabilities for carbon nucleation and structure deformation [21,22,23]. However, the transformations between a carbon sp^2^- and sp^3^-hybrid at the nanoscale are the most important processes which would govern the final structure’s stabilities and physical properties [24]. And the transition from sp^2^ to sp^3^ to form the high-concentration four-fold coordinated carbon is unanimously deemed to be the prerequisite for diamond nucleation [25,26,27,28,29,30,31,32].

Concretely, Caro et al. [26] investigated the origin of high sp^3^ content in tetrahedral amorphous carbon and proposed that high sp^3^ fractions arise from a delicate balance of transitions between three-fold and four-fold coordinated carbon atoms. Tao and coworkers [27] studied the sp^2^ to sp^3^ transition under pressure using Raman spectroscopy and suggested that water molecules can promote the formation of sp^3^-bonding conformation. Zhu et al. [28] reported a revisited mechanism of graphite–diamond transition at high temperatures and proposed that the diamond nucleus is more likely formed along the (1 2 0) and (0 0 1) direction of the graphite matrix. Nian et al. [33] studied the graphite to diamond phase transition using the nanosecond laser shock detonation technique and uncovered the critical transition temperature and pressure for the sp^2^- to sp^3^-hybrid. Billups and coworkers [29] revealed the radiation-induced nucleation of the diamond from amorphous carbon and estimated the phase transition pressure based on cluster size. In addition, the theoretical studies performed by Dong [30] and Liu et al. [31] displayed the different transition paths from graphite to diamond. They revealed the probable transition state of sp^2^ to sp^3^ and the facile nucleation mechanism of a hexagonal diamond. Despite that, the intrinsic atomic mechanism involved in carbon nucleation and microscopic structure evolution in explosives detonation is not clearly understood yet.

In this article, we investigate the formation and structure properties of condensed carbon materials formed in TATB detonation reaction, using the quantum-based molecular dynamics method. The structural evolution of initial carbon products is investigated at different pressures and temperatures close to the Chapman–Jouguet (CJ) state. The microscopic structure features are revealed by combined analysis of atom coordination number, molecular ring components, and bond angle distribution. Diverse carbon structures such as, graphene-like, diamond-like, and “diaphite”, are detected under different pressures. The typical mechanism of the C sp^2^- to sp^3^-hybrid transition is uncovered. The nucleation mechanism and structural characteristic of diamond-like carbon are revealed using radial pair distribution function (RDF), electronic density of states (DOS), and X-ray adsorption spectroscopy (XAS). The complex “diaphite” configuration is identified to be formed during the nucleation process. Our results give a new understanding of the nucleation of detonation diamond-like carbon, which would shed light on the synthesis of nano-carbon materials under high pressure.

## 2. Results and Discussion

In this study, we performed four dynamics simulations for 340 ps at different temperatures and pressures to reveal the dynamic deformation process of the carbon structure. Based on the gaseous products removal strategy, major heteroatoms of N/O/H were gradually released. The populations of main gaseous products are shown in Appendix A and the effects of temperature and pressure on the heteroatom’s liberation are illustrated there in detail. Here, we just focus on the nucleation process at 3500 K to uncover the microscopic features of the detonation condensed-phase carbon under different pressures.

### 2.1. Dynamic Structure Evolution of Carbon Clusters

The drastic structure deformations of the carbon cluster are displayed in Figure 1. All the initial clusters experience fast structure rearrangement during the first 40 ps because the chemical environment is significantly changed when they are separated from the detonation product system. Numerous small molecular fragments are produced with part of the cluster disaggregation (see the snapshots at 40 ps in Figure 1), which can easily collide and react with each other to form the products of CO_2_, N_2_, and H_2_O. As the gas molecules liberate, the large carbon-branched chains are constructed through translational motion, rotation, and the twisting of remaining molecular fragments. Some single molecular rings are generated in this process (see Appendix A), and even a few irregular polycyclics are formed under 50 and 60 GPa. This is similar to the graphene formation process reported by Wen et al. [34]. Further collision and aggregation can directly produce more polycyclic structures, resulting in the formation of small quasi-plane configurations (see the snapshots at 115 ps for 40 and 50 GPa, and 70 ps for 60 GPa in Figure 1 and Appendix A). Up to 190 ps, the initial plane structure is almost completed under 50 GPa, which is really ambiguous for the condition of 40 GPa. It illustrates that it is preferable for the initial carbon layers to be formed under higher pressure. Moreover, the higher pressure can promote the formation of quadruple-coordinated carbon atoms, with the bonding of dangling C atoms between adjacent molecular layers (such as the snapshots of 280 ps under 50 GPa, and 160 ps under 60 GPa). More significantly, we observe the obvious rearrangement phenomenon of the initial layer configuration from 70 to 160 ps under 60 GPa, and the disorder cluster configuration is generated temporarily (see the snapshot of 115 ps in Figure 1c). It illustrates that the carbon layer structure is not stable under such high temperatures and pressure.

Figure 2 shows the bonded features of carbon atoms during the structure deformation. The average coordination numbers of carbon increase gradually as the reaction proceeds, with the ranges from 1.925 to 2.893, 2.333 to 2.950, and 2.650 to 3.230 for the conditions of 40 GPa, 50 GPa, and 60 GPa, respectively. Obviously, the higher pressure can efficiently promote the formation of a more multi-coordinated configuration. Under 40, 50 GPa, the fraction of triple-bonded carbon atoms with a sp^2^-hybrid increase continuously, while that of the double- and single-bonded atoms decrease correspondingly. At the end of the simulation, almost all the single-bonded structures convert into multi-bonded structures, illustrating the high activity and low stability of the dangling atoms. A few numbers of double-bonded atoms remain in the system, whose number becomes smaller when yielding to the higher pressure of 50 GPa (see Figure 2a,b). It denotes that high pressure can promote the conversion from carbon sp- to a sp^2^-hybrid. Although some quadruple-bonded C atoms are observed during the dynamics process, they cannot exist stably, and almost no sp^3^-hybrid carbon remained after the simulation. As for 60 GPa, the fraction evolutions of double- and single-bonded carbon are similar to the conditions of 40, and 50 GPa, but the apparent transformations between triple- and quadruple-bonded structures (sp^2^↔sp^3^) are observed for the first time. This process is reversible and happens several times (see Figure 2c and Appendix A), representing a delicately dynamic equilibrium for these two conformations. After 220 ps, a slight advantage is observed for the conversion of sp^2^ to sp^3^, with the ratio of sp^3^ to sp^2^ keeping a relatively large value. And the final fraction of sp^3^-hybridized atoms reaches about 26%.

As the bonded structure deforms the carbon cluster, many carbon-based fragments can rearrange into various molecular rings. Figure 3 displays the population evolutions of carbon rings with different sizes. The cluster mainly consists of five, six, seven, and eight-member rings. All the molecular rings increase gradually at an early stage, which indicates that the degree of crystallization of carbon clusters is enhanced [35]. Among them, the hexatomic ring is the most stable unit, which possesses the highest formation rate and population. For the condition of 40 GPa, the formation rate of the pentatomic ring is almost comparable with that of the hexatomic ring at the first 115 ps. After a slight decrease, their populations keep a considerable constant during the following process. It illustrates that the formation of a five-member ring is also the favorite at such high temperatures and pressure. The populations of heptatomic and octatomic rings also display an increase at first and then a decrease, but their values are smaller than the other two. Moreover, a rapid formation of a hexatomic ring is observed after a temporary oscillation (from 190 ps shown in Figure 3a), indicating an absolute advantage for its formation in this stage. The evolution trends of different rings for 50 GPa are similar to that of 40 GPa, but the populations of heptatomic and octatomic rings are slightly larger than the latter (see Figure 3b).

As for 60 GPa (see Figure 3c), almost all of the rings possess larger populations than the other two conditions at the early stage, which demonstrates that the higher pressure is more beneficial for carbon cluster crystallization. The reversible transformation behaviors between hexatomic and non-hexatomic rings are observed. During the first 100 ps, almost all of the rings have considerable increments. After that, the hexatomic rings keep their increase, while the non-hexatomic rings display a significant decrease. It denotes that many non-hexatomic rings convert into hexatomic rings in this stage. And the reversed transition process occurs from ~205 ps, with a drastic decrease for the hexatomic rings. Thus, the equilibrium between different rings is temporary and delicate under 60 GPa. Such transformation phenomenon is consistent with the conversion of the carbon sp^2^-↔sp^3^-hybrid, which is more clearly confirmed by comparing the ratio evolution of sp^3^ to sp^2^ and non-hexagon to hexagon rings (see the condition of 60 GPa in Appendix A). It illustrates that the orbital hybrid of carbon has a close relationship with the formation of different rings during cluster deformation.

### 2.2. Final Configures of Carbon Cluster at Different Conditions

Figure 4 shows the final snapshots of the carbon structures at 3500 K, with element components of C_109_N_21_O_6_H_5_, C_111_N_23_O_9_H_3_, and C_112_N_23_O_12_H_5_ for 40, 50, and 60 GPa, respectively. It denotes that the purity of carbon clusters has a significant increase, with more than 81% N, 84% O, and 82% H atoms released, respectively. Under lower pressures of 40 and 50 GPa, the carbon clusters finally convert into typical graphene-like layer structures (see Figure 4a,b), with the remaining N/O/H heteroatoms implanting in. From the top view, oxygen atoms always locate on the margin or gap of the carbon layer to mainly form hydroxyl or carbonyl, which is consistent with the results observed in layered graphene formation [35]. In contrast, nitrogen atoms can indiscriminately distribute in the whole layer but are always accompanied by the formation of a non-hexatomic ring when they are located in the middle of the layer, especially for the condition of 50 GPa. The non-hexatomic rings are not usually isolated. The five-member ring always combines with the seven/eight-member rings to form multi-ring structures (see top view in Figure 4a,b). This result agrees well with the defects observed in graphene formation both in theoretical and experimental studies [36,37].

As for 60 GPa (see Figure 4c), the initially formed layer structure is totally destroyed and reconstructed into amorphous carbon with a high fraction of the sp^3^-hybrid (see Figure 2c). There are 9 five-member, 29 six-member, 8 seven-member, and 5 eight-member rings within the cluster. No significant difference is observed from the number of different rings, but a typical net carbon nucleus is formed in the middle of this cluster (see bright atoms in Figure 4c), with few N atoms embedding in.

The features of the final structures are further characterized by the radial pair distribution function (RDF, see Figure 4d). The primary peaks for conditions of 40 and 50 GPa are located in 1.41, 2.43, and 3.65 Å, which just correspond to the first, second, and fourth peaks of the C-C RDF of typical graphene (see Appendix A). The peak of 3.65 Å represents the interatomic position of adjacent rings, which indicates the formation of the honeycomb structure. However, the third peak (~2.76 Å) corresponding to the hexatomic ring [35] is overlapped with the second one. This is primarily due to the existence of many non-hexatomic rings in the cluster. The *g*_c-c_(r) RDF curve of 60 GPa displays four peaks at about 1.48, 2.48, 3.69, and 4.39 Å. They keep consistent with the first, second, fourth, and fifth peaks of the C-C RDF diamonds. It confirms that the carbon nucleus obtained in our study has some diamond structure features.

### 2.3. Transformation Mechanism of C sp^2^- to sp^3^-Hybrid under 60 GPa

From the bonded structure analysis above, we uncovered the carbon sp^2^↔sp^3^ transition process at 60 GPa, but the atomic mechanism relevant to the microscopic structure deformation was not clearly understood. Figure 5 further reveals the structure feature of the carbon cluster by analyzing the bond angle distributions of C-C-C. The main bond angle is changed between graphene-like and diamond-like structures reversibly. The typical conversion processes, such as 40–115 ps, 115–160 ps, and 220–295 ps, strictly correspond to the change of the bonded structure of carbon atoms (see Figure 2c). For simplification, we focused on the transformation stage from 220 ps to uncover the intrinsic transfer mechanism of the carbon sp^2^- to sp^3^-hybrid.

The graphene-like carbon layer with few nitrogen atoms is first formed at 220 ps (see Figure 6a), which indicates that the graphene-like carbon would be more readily constructed during an early detonation reaction. The carbon layer only consists of hexatomic rings and can steadily exist for more than 15 picoseconds, while, with vibration and collision of the dangling carbon atoms at the margin, some can overcome the energy barrier to form temporary quadruple-coordinated conformation, with C-sp^2^ transforming to C-sp^3^ (see purple atoms in Figure 6b). As a result, the conjugated π bonds structure within the six-member ring are destroyed, and the corresponding C-C bond rupture to form new stable conformations with 5 seven-member rings (see the ring transformation during 238.73 to 241.18 ps in Figure 6c). The carbon atoms within newly formed five- or seven-member rings obviously prefer to generate tetrahedral conformations with the sp^3^-hybrid (see purple atoms in Figure 6c). The initial carbon layer is further destroyed with the cross-bonding atoms out of the plane. Thus, the transition from a C sp^2^ to a sp^3^-hybrid is initiated by forming the delicate quadruple-coordinated structure, which directly induces the rings to transform from hexagon to pentagon or heptagon. The atoms within these non-hexagon rings acting as key sites can efficiently promote the conversion of a C sp^2^- to sp^3^-hybrid.

### 2.4. Diamond-Like Carbon Nucleation Mechanism and Microscopic Structure Characteristics

As the conversion of carbon sp^2^- to sp^3^-hybrid, many tetrahedral structures are produced. Some of them can aggregate together to form a diamond-like carbon nucleus. As seen in Figure 7, two main steps are revealed during the nucleation process: (I) the carbon atoms or fragments directly combine with the initial graphene-like layer (marked with a golden ball in Figure 7a). As further deforming of the graphene-like carbon occurs, the surrounding dangling carbon atoms can easily bond with the layer to produce tetrahedral structures. The new five-member rings are always preferred to be generated during this process (see the top view in Figure 7a), which can provide a more stable sp^3^-hybridized conformation for the carbon atoms. On the other hand, similar behavior also occurs between the other polycyclic carbon layer and dangling carbon atoms around it (see bottom view in Figure 7a). After that, (II), the significant aggregation between these layer structures is observed. As seen in Figure 7b, some new covalent bonds are formed between the adjacent carbon atoms at the boundary of these carbon structures. The more stable bonds are always formed between carbon atoms within pentagons, or within newly generated pentagons (see green bonds labeled in Figure 7c). It definitely identifies the importance of a five-member ring for diamond-like carbon nucleation. With further coagulation, the internal quasi-plane ring structures are totally destroyed, and a new carbon nucleus, mainly consisting of tetrahedral carbon, is formed (see light atoms in Figure 7d,e). Namely, the nucleation of detonation diamond-like carbon is dominated by the deformation and aggregation of graphene-like carbon layers, which is obviously different from the well-known mechanisms of BEN [19] or CVD [20] under lower temperatures and pressure.

It is worth noting that the left part of this nucleus is almost constructed by all the sp^3^-hybrid carbon atoms (see Figure 7e-II), which contain more than 36.6% C atoms of the whole cluster. The center atoms marked in purple color have the typical diamond-like structure, constructed with several non-planar hexagon rings (labeled with pink sticks). The electronic density of states (DOS) for the corresponding carbon structure is calculated using the density functional theory method [38] and shown in Figure 8a. There are considerable electronic states across the Fermi level, which indicates the total system is the metallic state. It is similar to the results observed in the structure study for amorphous carbon [39]. However, almost no contribution of the diamond-like carbon nucleus around the Fermi level is observed. The local DOS of diamond-like carbon atoms shows a clear energy gap of ~5 eV, which is comparable with that of diamond (~5.5 eV) [19,40], further confirming the diamond-like features of this carbon nucleus.

Furthermore, three layer-like structures are formed, combining with the right margin of the diamond-like nucleus, with a dihedral angle of about 50 degrees (see Figure 8b). It is similar to the interface structure of “diaphite” observed during the synthesis of a diamond–graphite composite in recent research [41,42]. Figure 8c shows the X-ray adsorption spectroscopy (XAS) of total carbon within the cluster, diamond-like carbon within the center nucleus, and “diaphite” carbon localized at the margin between the diamond-like nucleus and graphene-like layers, respectively. Although the sp^2^-hybrid dominates the cluster structure as shown in Figure 2c, only a slight signal of K-edge XAS around 280 eV (C-1s→π*) is detected for total carbon, while a strong σ* peak appears at ~292 eV (C-1s→σ*). This result keeps consistent with that of the amorphous carbon simulated in the previous study [43]. It illustrates that no considerable π bond structures remain in the carbon cluster and the whole system is still disordered to some degree. The spectra for center diamond-like carbon show a clear sp^3^ signal at about 292 eV, without π* peak detected. It agrees well with the K-edge of diamond obtained in relevant experiments [42,44], further identifying its diamond-like features. As for the carbon atoms at the diamond–graphite margin, the XAS spectra are dominated by the σ* peak, with a small shoulder peak corresponding to the excitation to π* orbital (~278 eV). Obviously, the margin atoms possess both the sp^2^ and sp^3^-hybrid properties, which would further grow to form the diamond–graphite interface and affect the properties of detonation nano-diamond [45]. Although Németh et al. [45,46] reported that the graphite-structured nano-carbons are widespread in nano-diamond, our results are the first to confirm that the complex “diaphite” structure has already been formed during the nucleation of detonation nano-diamond. Namely, the initial diamond-like nucleus is surrounded by graphene-like carbon layers, and the following growth of diamond-like carbon would accompany the intergrowth of graphene-like carbon [47]. It needs to be mentioned that our simulated π* peaks have a slight left shift compared with the electron energy loss spectroscopy (EELS) experimental results [42]. It mainly ascribes to that the non-planar ring is not the stable π-conjugate configuration and it is always implanted in some N heteroatoms (see the stick structure in Figure 8b). As a result, the formed π bond orbitals have a higher energy than the typical structure, with their corresponding anti-bond orbitals (π*) locating a lower energy level.

**Figure 8 ijms-24-12568-f008:**
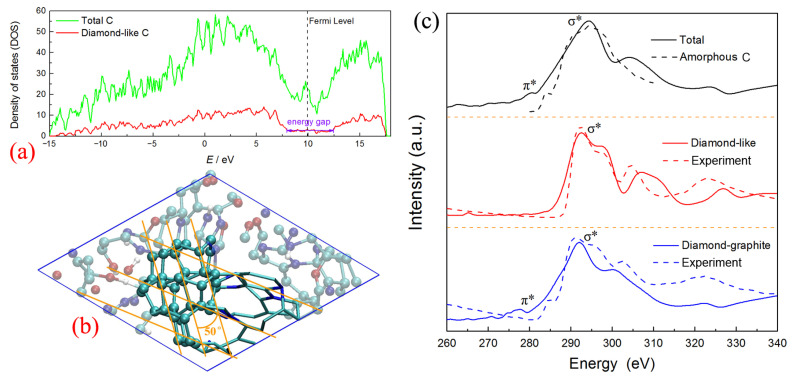
The structure features for diamond-like carbon nucleus: (**a**) density of states, (**b**) )snapshot of diamond–graphite configuration, (**c**) X-ray adsorption spectroscopy (theoretical calculation for amorphous carbon [43], experimental result for diamond-like and diamond-graphite carbon [42]; further permissions related to the materials excerpted here should be directed to the original journals).

## 3. Materials and Methods

The dynamic structure evolution of the carbon cluster from the TATB detonation reaction is simulated using the self-consistent charge density functional tight-binding (SCC-DFTB) [48] method implemented in the CP2K [49] code. It is based on the second-order expansion of the Kohn–Sham total energy in density functional theory with respect to charge density fluctuations, which allows one to describe the total energies, atomic forces, and charge transfer in a self-consistent manner. Its reliability has already been confirmed in previous studies on the description of microscopic structures, electron properties, and reaction behaviors of commonly energetic materials [50,51,52,53,54].

The initial condensed-phase carbon structure is drawn from the shock decomposition of TATB reported in previous work [55]. The total detonation products at 150 ps are displayed in Figure 9a,b, and the heteroatomic cluster is directly separated from the product system (shown in Figure 9c), with a chemical component of C_138_N_125_O_76_H_29_. The initial lattice parameters are *a* = 9.264 Å, *b* = 18.001 Å, *c* = 17.993 Å; *α* = 119.93°, *β* = *γ* = 90°. The dynamics evolution process is carried out using *NPT* canonical ensemble. A NOSE thermostat [56] and a new barostat proposed by Martyna et al. [57] are used to keep temperature and pressure at aim values, respectively. Under the typical detonation temperature of 3500 K, both the near and over-detonation pressures of 40, 50, and 60 GPa are investigated. Because only one hundred carbon atoms are involved in this study, higher pressures are employed to ensure the phase transition from graphite to diamond, according to the phase diagram based on hydrodynamic computation [10]. An additional condition with a low temperature of 3000 K and a pressure of 50 GPa is considered for comparison. The initial cluster structures are first relaxed at different conditions for 40 ps. Thereafter, the additional 300 ps are performed to simulate the dynamic evolution process of condensed-phase carbon. The dynamics time step is set to 0.5 fs, with a self-consistent field (SCF) convergence of 10^−6^ au.

The X-ray adsorption spectroscopy (XAS) for typical carbon structures is calculated using linear-response time-dependent density functional theory (LR-TDDFT) [58] also implemented in CP2K code. The Gaussian and augmented plane wave (GAPW) [59] method is employed for an accurate description of core states. The exchange–correlation interaction is described by the Perdew–Burke–Ernzerhof (PBE) [60] function, with a 6-311G** [61] basis set. The plane wave cutoff energy is 600 Ry. The SCF convergence value is set to 10^−6^ au. The C 1 s state is selected to be excited during the XAS simulation, corresponding to the K-edge. The obtained excitation energy and oscillator strengths data are processed based on Gaussian extension to match the experimental results.

The MD trajectories are systematically analyzed using an MD post-processing code compiled by fortran90. The stable chemical bonds and molecular components are identified according to bond length and lifetime criteria [62,63]. The structure features of carbon clusters are analyzed using RINGS code [64], which is the rigorous investigation of networks generated in MD simulations. It is a scientific code developed in Fortran90/MPI to analyze atomic connectivity using ring statistics. A stable gaseous products (H_2_O, CO_2_, N_2_) removal strategy is employed every 15 ps to accelerate the heteroatom liberation and improve the purity of the carbon cluster (more details illustrated in Appendix A).

## 4. Conclusions

In summary, using the quantum-based molecular dynamics method, we reveal the structure dynamic evolution process of condensed-phase carbon formed in TATB detonation. The graphene-like layer configurations are obtained under low pressures of 40 and 50 GPa, while the typical diamond-like carbon nucleus is observed under high pressure of 60 GPa. The transition mechanism of C sp^2^- to sp^3^-hybrid is uncovered under high pressure. The graphene-like layers are favorably formed at the early stage. As the four-fold coordinated conformation temporarily forms between the dangling carbon atoms at the margin of the layers, the conjugate π bond interaction is destroyed. It is beneficial for the conversion from a hexatomic ring to a non-hexatomic ring and further promotes the formation of high-concentration sp^3^-hybridized carbon. The diamond-like carbon nucleus is further constructed by aggregation of the deformed graphene-like layers, with its margin surrounded by graphite-like carbon to form the “diaphite” configuration. It clearly indicates that the nucleation and growth of detonation nano-diamond are tightly bounded with the graphene-like carbon layers, which is obviously different from the previous mechanisms of BEN or CVD under lower temperatures and pressure. Thus, the nano-diamond or ‘diaphite’ complex materials with special structural and physical properties may be synthesized by controlling the structure characteristic of initially formed graphene-like carbon. Our study provides a new understanding of the formation and structure characteristics of detonation product carbon, which may give significant inspiration for the experimental synthesis of nano-carbon materials through explosives detonation.

## Figures and Tables

**Figure 1 ijms-24-12568-f001:**
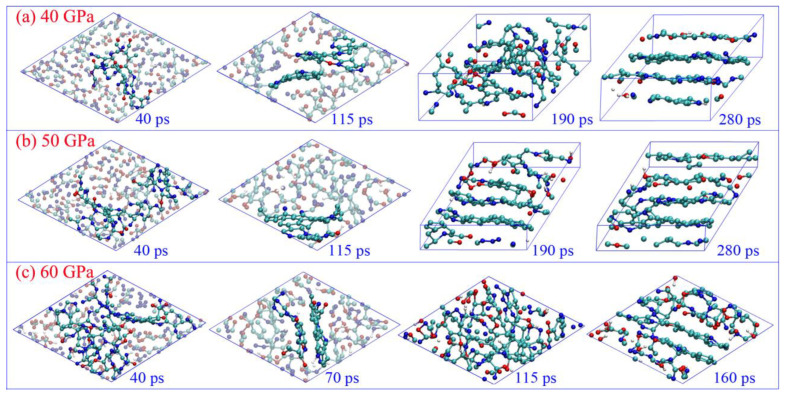
Time evolution of the microscopic structure of carbon cluster under different pressures (cyan ball for carbon, blue ball for nitrogen, red ball for oxygen, and white ball for hydrogen).

**Figure 2 ijms-24-12568-f002:**
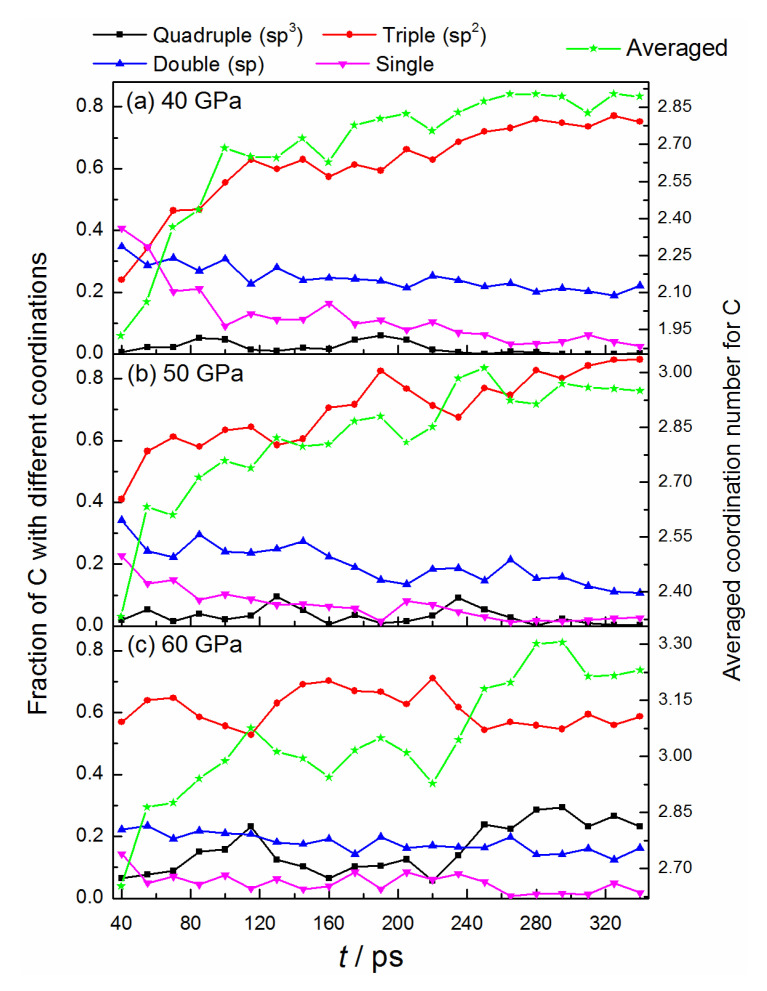
Time evolution of bonded structures of carbon under different pressures.

**Figure 3 ijms-24-12568-f003:**
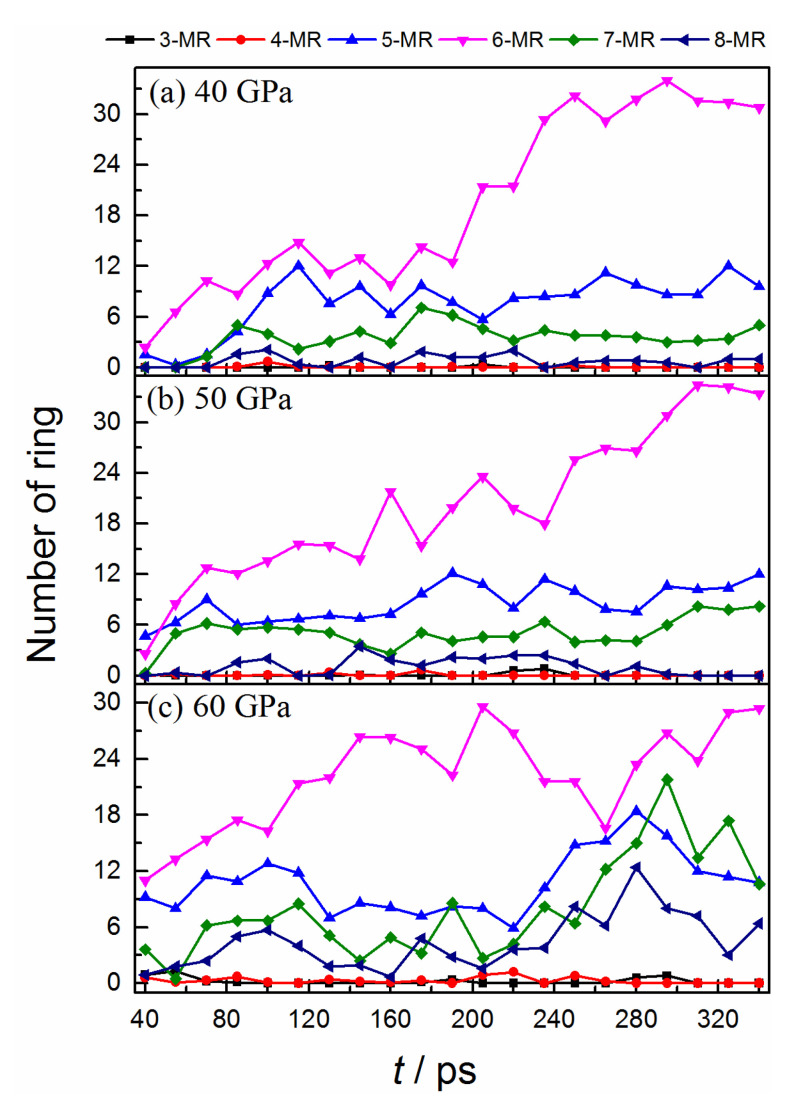
Time evolution of the numbers of different molecular rings.

**Figure 4 ijms-24-12568-f004:**
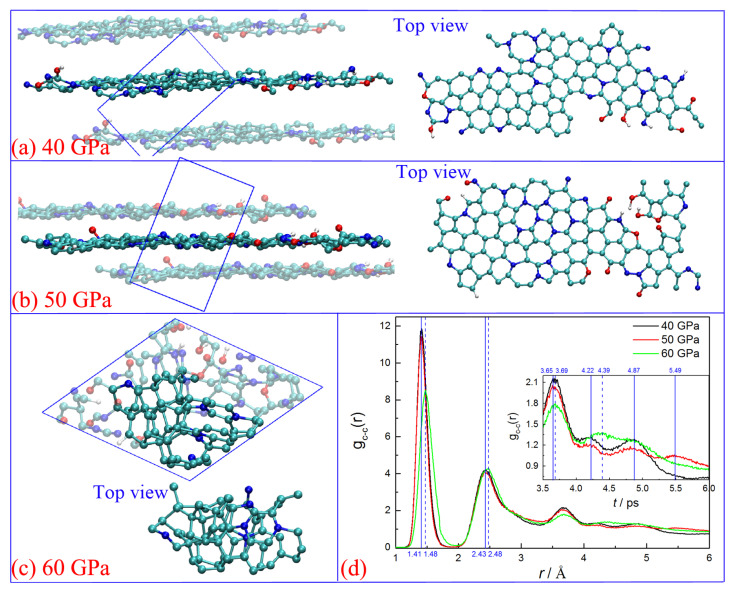
Snapshots for the final configurations (**a**–**c**) and radical pair distribution function of C-C (**d**) under different pressures, (cyan ball for carbon, blue ball for nitrogen, red ball for oxygen, and white ball for hydrogen).

**Figure 5 ijms-24-12568-f005:**
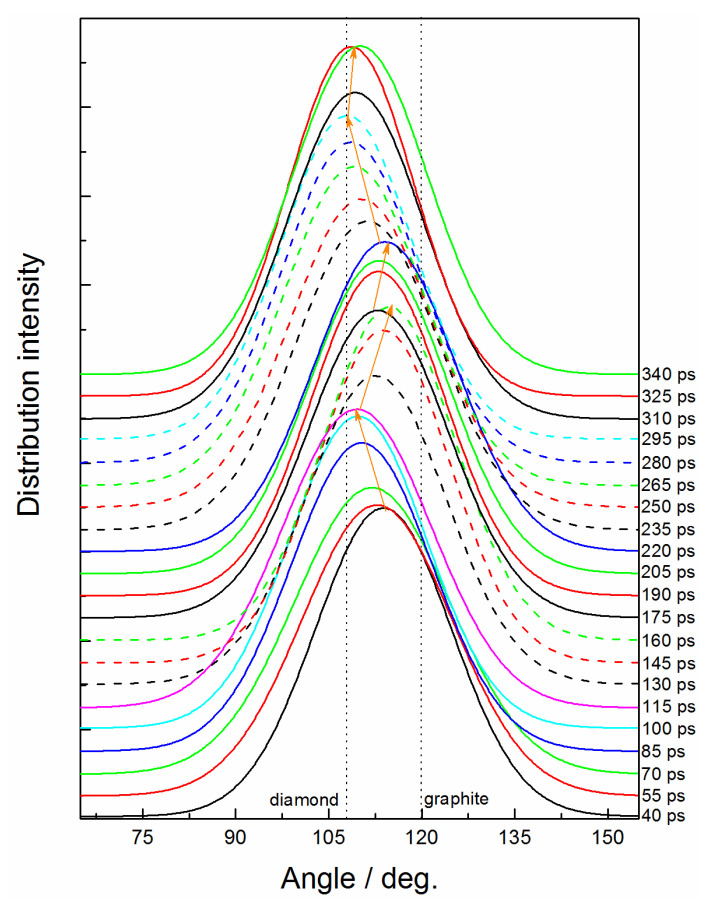
Time evolution of bond angle for C-C-C under 60 GPa.

**Figure 6 ijms-24-12568-f006:**
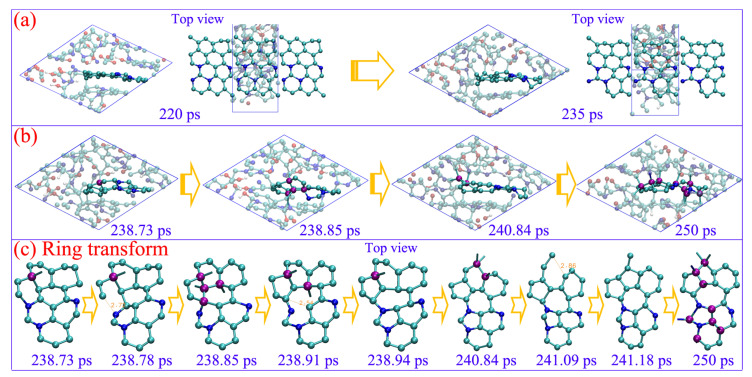
The carbon sp^2^–sp^3^ transformation (**a**,**b**) and corresponding ring structure evolutions (**b**) under 60 GPa (cyan ball for carbon, blue ball for nitrogen, red ball for oxygen, and white ball for hydrogen).

**Figure 7 ijms-24-12568-f007:**
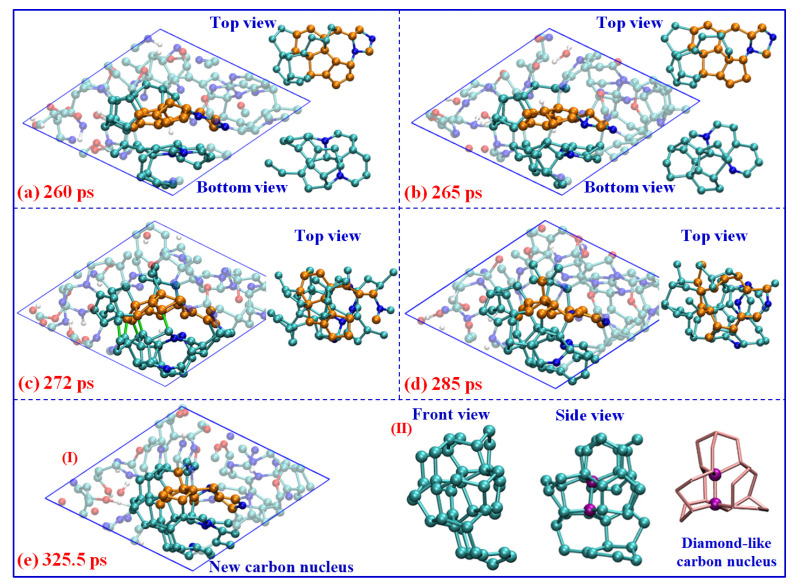
Nucleation of the diamond-like carbon under 60 GPa (cyan ball for carbon, blue ball for nitrogen, red ball for oxygen, and white ball for hydrogen).

**Figure 9 ijms-24-12568-f009:**
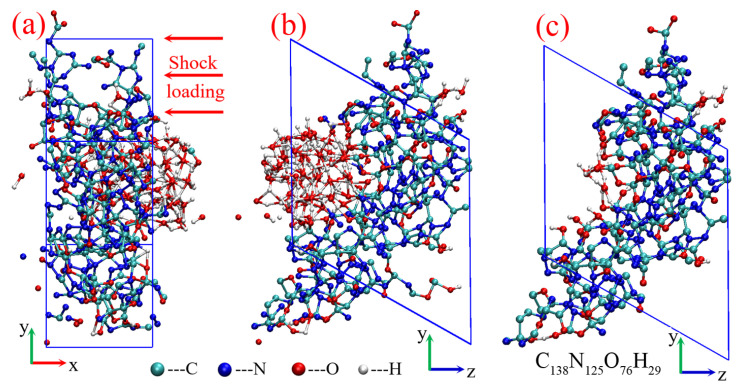
Initial cluster model from the TATB shock decomposition: (**a**,**b**) product system of TATB, (**c**) initial cluster structure.

## Data Availability

The datasets analyzed during this current study are available from the corresponding author upon reasonable request.

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
