# Peer review of "The Structure Properties of Carbon Materials Formed in 2,4,6-Triamino-1,3,5-Trinitrobenzene Detonation: A Theoretical Insight for Nucleation of Diamond-like Carbon"

_ijms, 2023, doi:10.3390/ijms241612568_

Round 1

Reviewer 1 Report

Building a comprehensive understanding of diamond nanostructure formation (DND) during a high explosive detonation is a challenging task. The complex and highly dynamic nature of the detonation process makes it difficult to perform in situ characterization on the sub-microsecond time-scale, which is necessary to capture the rapid formation and evolution of these nanostructures.

The authors aim to address the lack of understanding regarding the nucleation mechanism and microscopic structure characteristics of diamond-like carbon.

The transition from C sp2- to sp3-hybridization, driven by the conversion of hexatonic to non-hexatomic rings, is detected under high pressure.

By employing a quantum-based molecular dynamics method, the authors conduct in-depth investigations into the dynamic evolution of condensed-phase carbon during TATB detonation.

The paper highlights the significance of comprehending the nucleation mechanism and microscopic structure characteristics in the design and production of carbon materials formed in explosive detonations.

The nucleation mechanism for diamond-like carbon revealed in this study, involving the direct combination of carbon atoms or fragments with graphene-like layers, provides vital insights into the growth process.

Overall, this article expands our understanding of detonation diamond-like carbon nucleation and offers important implications for the synthesis and design of nano-carbon materials under high pressure.

However, authors should explain, how specifically proposed mechanisms can be used to program the nucleation process at synthesis of nano-carbon materials through explosives detonation?

The structure of the paper should be modified. The quantum-based molecular dynamics method used should be described in more detail and placed in a section before the simulation results, i.e. section 3 should be placed before section 2.

Can the authors provide a link to the CP2K code?

Since the Abstract and Conclusion sections in present form are not informative enough, I suggest here include the key microscopic features of the suggested nucleation mechanism for diamond-like carbon as well as the novelty of this mechanism in comparison with previous research.

The bibliographic list should be supplemented by the following publications, connected with mechanisms of carbon phases transformations:

- Vejpravová, J. Mixed sp2–sp3 Nanocarbon Materials: A Status Quo Review. Nanomaterials 2021, 11, 2469. https://doi.org/10.3390/nano11102469

- Shang, S.-Y.; Tong, Y.; Wang, Z.-C.; Huang, F.-L. Study on the Polycrystalline Mechanism of Polycrystalline Diamond Synthesized from Graphite by Direct Detonation Method. Materials 2022, 15, 4154. https://doi.org/10.3390/ma15124154

- Joshua A. Hammons, Michael H. Nielsen, Michael Bagge-Hansen, Sorin Bastea, William L. Shaw, Jonathan R. I. Lee, Jan Ilavsky, Nicholas Sinclair, Kamel Fezzaa, Lisa M. Lauderbach, Ralph L. Hodgin, Daniel A. Orlikowski, Laurence E. Fried, and Trevor M. Willey, Resolving Detonation Nanodiamond Size Evolution and Morphology at Sub-Microsecond Timescales during High-Explosive Detonations, The Journal of Physical Chemistry C 2019 123 (31), 19153-19164, DOI: 10.1021/acs.jpcc.9b02692

- Xing Zhang et al, Nanosecond laser shock detonation of nanodiamonds: from laser-matter interaction to graphite-to-diamond phase transition, 2022 Int. J. Extrem. Manuf. 4 015401, https://doi.org/10.1088/2631-7990/ac37f1

All bibliographic references must also contain DOI numbers.

The references #45 and #59 should be corrected to match the MDPI style.

Authors must include a List of Abbreviations in the paper.

The abbreviation (TATB) used in the title is not deciphered anywhere in the abstract and in the text of the paper. The use of an abbreviation in the title of the paper is not allowed.

There are also other abbreviations that need to be deciphered in the paper: “TNT”, line 38; "CJ state", line 71; “CP2K”, line 310; “PBE”, line 337, “SCF”, line 338, “6-311G**”, line 338.

The quality of the Figs. 1, 4, 6-8 and 9 must be improved.

The paper cannot be published in its present form and requires significant improvements.

Thank you!

Reviewer 2 Report

The whole draft should be carefully proofread to ensure there are no grammatical errors or typos that could affect the overall readability and professionalism of the manuscript.

Reviewer 3 Report

He et al. theoretically studied the structure properties of carbon materials and the structure dynamical evolution process of condensed-phase carbon formed in TATB detonation using the quantum-based molecular dynamics method.  The authors obtained graphene-like layer configurations under low pressures of 40 and 50 GPa. However, at high pressure of 60 GPa, the sp2 to sp3 transition occurs, and as a result, the carbon materials become typical diamond-like carbon. The study provides a new understanding of detonation product carbon's formation and structure characteristics. There are a few typographical errors that the authors want to revisit. The manuscript is written well and easy to follow.

There are a few typographical errors that the authors want to revisit.

Author Response

There are a few typographical errors that the authors want to revisit. The manuscript is written well and easy to follow.

Comments on the Quality of English Language

There are a few typographical errors that the authors want to revisit.

Answer: Thank you. We have carefully revised the whole manuscript, and give some necessary correction. The English has some improvements.

Round 2

Reviewer 1 Report

The revised paper looks much better.

Authors must include a list of Abbreviations in the paper.

The abbreviation "CJ" on line 74 must be placed in brackets.

The abbreviation “6-311G” in line 118 should be explained.

The text of the paper should include a brief description of the “CP2K code”, since readers may not be familiar with this code.

The same goes for the reference to "RINGS code" in line 126.

Again, can the authors provide a link to the CP2K code?

Reference [001] in line 61 looks erroneous and needs to be corrected.

All bibliographic references must also contain DOI numbers.

The Conclusion section should not contain references to publications, (BEN[19] or CVD[20], line 381), since such references can be given in the main section of the paper.

The quality of the Fig. 7 must be improved.

Thank you!

Reviewer 2 Report

Most of the questions and issues have been addressed, hence I can recommend publication.

The English language has seen some improvements.

Author Response

Thank you! We have carefully revised our manuscript.